# The Role of Parental Support and the Students’ Opinions in Active Finnish Physical Education Homework

**DOI:** 10.3390/ijerph191911924

**Published:** 2022-09-21

**Authors:** Mari Kääpä, Sanna Palomäki, Alicia Fedewa, Ulla Maija Valleala, Mirja Hirvensalo

**Affiliations:** 1Faculty of Sport and Health Science, University of Jyväskylä, P.O. Box 35, FI-40014 Jyväskylä, Finland; 2Educational, School and Counseling Psykology, University of Kentucky, Lexington, KY 40506, USA; 3Faculty of Education and Psychology, University of Jyväskylä, P.O. Box 35, FI-40014 Jyväskylä, Finland

**Keywords:** physical activity, adolescent, physical education homework, family support

## Abstract

Prior research indicates that adolescent boys are often more active than girls, implying a need for special attention to increase the physical activity levels of adolescent girls. Adolescents are at an age where they are especially susceptible to environmental and social influences but still have a limited amount of autonomy over their own behaviors. The effective physical activity programs implemented at this age may benefit health into adulthood. The fact that adolescents’ physical activity is influenced by many factors indicates that to achieve any behavioral change, interventions must target several levels across the socio-ecological model. During childhood, the family is the primary factor in socializing and shaping engagement in physical activity. This study is part of the Physical Education (PE) Homework Study project which was implemented in a midsized secondary school in the middle of Finland from 2016 to 2020. The goal was to develop one easily approachable way to prevent the decreasing physical activity of adolescent girls. This was done by increasing physical activity times of adolescent girls outside of the school by giving them active PE assignments. The aim was also to explore students’ and their parents’ perceptions of physically active physical education homework. In this part of the study, there were 43 interviews: 38 student interviews and 5 interviews with parents. The analysis process followed the qualitative content analysis (QCA) strategy by Schreirer. In this study, we combined the views of students and parents, and obtained a broad picture of the PE homework assignments given at school but completed at home. According to students and parents, PE homework assignments should be diverse, interesting, and challenging, they should also be provided at flexible schedules outside of school hours with family support. Physical education homework could be a potential approach to influence the physical activity of the student population by involving school curriculum and families.

## 1. Introduction

The future patterns of adult health are established during childhood [1,2,3]. For that reason, adolescents are an important target for health promotion, including physical activity (PA) [3,4,5]. Adolescents are at an age where they are especially susceptible to environmental and social influences but have a limited amount of autonomy over their own behaviors [4]. However, effective PA programs implemented at this age may benefit health into adulthood because previous physical activity is related to physical activity later in life [1,2,3]. Research indicates that adolescent boys are often more active than girls, implying a particular need to increase the physical activity levels of adolescent girls [1,2,6,7,8,9]. The findings of Dias et al. [6] support the notion that perceived barriers to physical activity during leisure time are more prevalent among girls. Furthermore, prior studies suggest that girls are more influenced than boys by different types of family support for physical activity [8,10] and therefore, if there is little family support for physical activity, girls are more likely to be adversely impacted in their lack of participation in sports and other physical activities [8,10].

Promoting physical activity in adolescence is a challenge because of the variety of social contexts in which they are a part as well as the reinforcing factors for what make some adolescents more active than others [9]. Silva et al. [9] found several variables to mediate the level of MVPA in girls: self-efficacy, peer social support, total social support, and difficulty getting to and from community activities. The socio-ecological model is a multilevel framework that considers the multitude of influencing layers on an individual’s behavior. This framework is often chosen as a guiding theory in health behavior studies [2,4,11,12,13] as it considers the biological, social, environmental, and even the public policy or organizational levels of influence [4,11,14,15]. The physical activity interventions that target the personal, interpersonal, and environmental domains of an individual are more likely to be effective [4,11].

### 1.1. Family Influence on Adolescent Physical Activity

The review by Martins et al. [5] concerning physical activity studies presented three key characteristics of physically active adolescents: high intrinsic motivation, social support, and environmental opportunities. Social support or family support has been consistently associated with adolescent physical activity [2,3,4,7,9,16,17,18,19], and it continues to be an important predictor of physical activity during teenage years [8]. Henriksen et al. [20] found that all measured parental social support items (encouragement, joining, watching, and talking about physical activity) were associated with adolescents’ physical activity levels. During childhood, the family is the primary factor in socializing and shaping engagement in physical activity and learning physical activity related habits, values, and beliefs [3,4]. According to Atkin et al. [3], positive family relations reduced the time spent in sedentary behaviors that are generally performed alone, such as reading, playing video games, or doing homework. Parental support had significant association with enjoyment, self-efficacy, and a direct effect on the level of MVPA in the study of adolescent physical activity by Silva et al. [9].

Several reviews summarize the research evidence related to parental influence on adolescents’ physical activity [2,19,21]. Trost and Loprinzi [19] identified four different types of parental influences: parental modelling, parental support, parenting style, and family cohesion, that demonstrated a significant positive association with adolescents’ physical activity when combined [19]. Sallis et al.’s [2] review of correlates of adolescents’ physical activity reinforces these findings as they indicate that parental support and direct help from parents were consistently related to adolescent physical activity. According to them, parental support can be verbal or direct assistance [2]. Heitzler et al. [20] concluded that role modelling of physical activity is more influential compared to other supportive behaviors by parents, suggesting that parents still play important roles in their teens’ lives. Biddle et al.’s [17] review calls for distinction between the parental support and parental physical activity. Their review implied that parental support comes in many different forms, which requires more unpacking and research [17].

Despite the importance of familial influence in physical activity, adolescence is also a period when peers become more central to the socialization process as the adolescent begins to individuate from their family [1,20]. This transition often contains behavioral risk factors for decreasing physical activity, especially among girls [4,17]. At earlier stages of adolescence, parents and family are more important, but as they become older, high levels of peer socializing increase adolescents’ odds of being active and peers’ influences may supersede parents’ influences over time [1,3,9,20]. In a Finnish study from Palomäki et al. [21], it was found that younger and more physically active youth perceived greater support from their parents. However, both the amount of support by the parents and its association with physical activity decreased with increased independence and autonomy due to aging and maturation in adolescence [21]. Thus, as adolescents become more independent with maturation, it will be important to draw on other sources of support and ways to contribute to increasing their levels of physical activity.

### 1.2. School-Level Efforts to Increase Adolescent Physical Activity

A growing number of school-based efforts address declining physical activity, even if the effect of policy on physical activity participation in schools is mixed [17]. Henriksen et al. [10] suggested that the school setting is a crucial factor when promoting participation in physical activities. A school-level approach reaches a high number of adolescents regardless of their social differences, creating a widespread population sample [4,11,22].

The Finnish National Curriculum, which is followed throughout the country, defines the boundary conditions and learning objectives for physical education [23]. All the teachers have university master’s level education, classroom teachers operate in grades from 1 to 6, and subject teachers at the lower secondary school (grades 7–9). In Finland, the distribution of physical education lesson hours during lower secondary school is seven hours weekly per year for three school years, meaning two or three hours per week during grades 7, 8, and 9. The most common way is to organize 90 min (2 × 45 min) of physical education weekly. In addition, students have the opportunity to choose optional physical education which adds weekly amounts of physical education. The school day often consists of 45-min lessons and 15-min breaks, which provides opportunities to be active during recess as well.

School sports contribute only marginally to adolescent’s physical activity, even well-organized physical education lessons do not exert a sufficient influence over physical activity and the health of adolescents [24,25,26,27]. To meet adequate levels of total physical activity, adolescents need to participate in physical activity outside of the school [28], the Finnish National Core Curriculum allows and encourages the use of leisure time to practice skills learned at school [23]. One strength of the Finnish educational system is that teachers have a great deal of autonomy in deciding the way they teach [29]. Teachers can create opportunities for decision-making and regulate the amount of student involvement in it. Teachers can improve students’ autonomy and self-determination by involving them in planning and decision-making, choosing scale and complexity, or amount of physical active assignments [30]. By also fostering perceptions of ownership over participation in physical education and physical activity, students are enabled to be active in a way they choose [30,31,32].

Increased PA is an integral component in enhancing school children’s physical health and wellbeing [33]. Even if effectiveness of school-based interventions is contradictory, it has been found that multicomponent school-based interventions have a positive effect on adolescent girls’ PA [34,35]. School-based PA can be overlooked despite its’ great potential and the multidimensional benefits for enhancing more systemic-level and sustainable change. In Finland, there are encouraging results of school-based intervention targeting physical activity during the school day [36]. Haapala et al.’s [36] large school-based intervention focused on spending recess outdoors, more organized recess activities, the greater provision of activity-based equipment, the development of sports facilities and applying gender-specific activities especially for girls, and producing a significant impact on youth PA [36]. Abdelghaffar et al. [4] used a slightly different school-based approach and found that the use of single-gender groups and involvement of parents and teachers proved to be a useful approach for addressing inactivity in girls [4]. In addition, Cook-Cottone et al.’s [37] study indicates that effective school-based PA intervention should include parents and concentrate on reduction of sedentary behavior [37]. Mears and Jago [35] meta-analysis indicates that after-school programs may offer an opportunity to increase children’s PA and reduce physical inactivity [35].

### 1.3. The Role of PE Homework

Integrating physical activity into the school day is one component to improve PA [36,38], but providing PE lesson activities into the home environment is not a typical approach in schools. Active PE homework assignments are usually related to PE curriculum and intend to add skill practice or preparation for PE lesson [39,40,41]. Studies recommend homework assignments that are versatile and offer students options to choose from [42,43]. Curriculum-based PE homework assignments could be comprised of basic motor skill practices, such as muscular training, balance practices, and endurance tasks, such as jogging or trekking [44,45,46], which are given to students during PE lessons by their PE teacher. PE homework can be done without any equipment, and research has shown that participants prefer doing them at home or outside of the school context [44]. Safe performance techniques and the skills required for PE homework are practiced at school. Assignments sometimes require the participation of others, such as family members or friends (e.g., teaching the squat technique to a family member). Active PE homework assignments are rarely reported in the research literature, but some studies have focused on active PE homework assignments [39,40,41,42,43,47]. One of the studies that have included PE homework is the Active Schools Strategies among sixth-to-eight grade children in Wisconsin [11]. In Bowser’s [11] project, homework involving physical activity was one of the strategies that had a significant level of reach, and active PE homework was one of the strategies most likely to be continued. The reason for the success of this part of the strategy was that active PE homework did not require any modifications to the school day, no additional time by teachers outside of the PE lessons was needed, and assignments were done without any equipment from the school [11].

### 1.4. Purpose

Given the success of PE homework, few studies have empirically validated its effectiveness, the aim of this study was to explore adolescents’ and parents’ perceptions of PE homework and to obtain a view of female students’ and parents’ opinions regarding performing PE homework. This study is part of the Finnish PE Homework Study, which addresses influencing factors in the context of the socio-ecological model [9,10] and adds to the knowledge and understanding of these interactions influencing PE homework and adolescent PA. This part of the study focuses on individual and immediate community levels and interaction between those levels (Figure 1).

## 2. Materials and Methods

### 2.1. Procedures

This study is part of the PE Homework Study project which was implemented in a midsized secondary school in the middle of Finland from 2016 to 2020. The goal of the study project was an attempt to increase PA time of adolescent girls outside of the school [44,45,46]. PE homework assignments were offered to all female students in this school but participating in the study was optional. Of 124 girl students, 105 girls agreed to participate in the PE Homework Study project, and 88 delivered acceptable data from accelerometers and diaries, during the study week. The lower secondary school students were from grades 7 to 9, and aged from 13 to 15 years old.

In this part of the study, there were 43 interviews: 38 student interviews and 5 interviews with parents. Of the five parent interviews, two parents had one child and three of the participants had two children in the school participating in the study. Four of the parent interviewees were female and one was male, and interviewees were selected by asking volunteers via the students’ parent mail list. Parents of student participants and adult participants were informed about the research and the Ethical Board instructions and procedures, their consent was asked for and ensured. The layer of the University Ethical Board approved these measures to be sufficient prior to the onset of the study.

At the beginning of the study—before providing the PE homework assignments—the parents were informed about PE homework practice via the parents’ intranet. PE homework assignments in this study followed the PE curriculum and were versatile. PE homework assignments were given once a week during the entire school year. Students had an opportunity to participate in decision-making by selecting skill level and intensity of their tasks. Including adolescents’ ideas in the activities and incorporating adolescents into the decision-making helped to encourage their motivation for completing the tasks [50].

The framework for the entire Finnish PE Homework Study is a socio-ecological model [12,48,49], which guided this part of the study as well. The focus was on students’ and parents’ perceptions about physically active PE homework, which broadens the study into individual, immediate community, and physical environment levels, including interaction between the levels. This socio-ecological model created the basis of the framework, and therefore categories of the coding frame were preconceived and thus deductive.

Student interviews were implemented during the spring of 2016. At this time, they were allocated PE homework assignments regularly for approximately six months. The voluntary students were asked to come to the dressing room for the interview, while a PE lesson by another teacher was taking place in the sport hall. The door to the hall was open, which allowed students to see their peers and ensuring the feeling of a safe environment. However, due to that, some sportive activities were recorded with interviews as well. Some of the students wanted to come in pairs, which was allowed. Group interviews could have been an option as well, but single or pair interviews assured that everyone had a chance to make their voices heard. Student interviews were on average under 5 min, and mean interview time was 3 min 20 s. The interview frame included issues concerning PE homework and interaction with parents, with whom and where PE homework were performed, barriers and facilitators in doing PE homework assignments, and experiences of their participation with the assignments.

The parent interviews took place during the spring of 2018. Three of the parent interviews were performed at interviewees’ homes, one was performed in the cafeteria during the lunch break, and one took place at an office after working hours—the mean interview time was 14 min. The interview frame followed the issues concerning the family’s sportive background and attitude towards PE homework, interaction with their children regarding PE homework, family support towards PE homework, and future wishes regarding PE homework. All the interviews were recorded and transcribed; originally interviews were conducted in Finnish. Due to that, the quotations are not verbatim, but they were translated as precisely as possible in an attempt to accurately convey the message of the speaker. The names have been changed to maintain anonymity.

### 2.2. Qualitative Content Analysis (QCA)

Qualitative content analysis (QCA) was selected because of the nature of the research questions, interview material, need to interpret, and its systematic feature, flexibility, and context involvement. The process followed the QCA strategy by Schreirer [51].

The focus of analysis was on family support of PE homework, which was the dimension that was used to create the main categories of the coding frame. Because of the data originating from two different sources, the decision to break the data down by source was made. As a teacher-researcher (first author), there is a possibility of being biased towards the material. Thus, there were two other persons doing the student interviews coding, which added the reliability by providing unprejudiced perspectives and alternative viewpoints of the data, as well as helping to correct the bias. Comparison across persons was done independently (blind coding) to ensure that analysis was intersubjective, and the categories apply across persons.

***Students’ interviews coding***—At first, all coders got to know the data and discussed them to ensure that everyone understood the categories the same way. Then coders collaborated to make the decisions about the coding frame and to describe and name categories. The subcategories were developed using the data-driven strategy of subsumption. After revising the data, they were divided into three parts for deeper evaluation. The definitions and indicators were made clear enough to avoid conceptual overlaps between categories (Table 1). The controversial parts of trial coding were solved by discussing afterwards, which clarified the criterion and coding frame.

***Parents’ interviews coding***—The parent interviews were coded and analyzed by one person. For reliability, material was compared across points in time. For the second time, the coding frame, categories, and indicators underwent an extra revision to ensure that the adjustments were correct and made the coding frame easier to understand and to use. There were only slight adjustments to the coding frame, for example, one subcategory was divided into two separate categories due to overlapping (Table 2).

The results are presented in quantitative terms by categories, not by cases. Especially younger students’ interviews were short and their answers quite simple, exploring qualitative data according to Schreirer’s [51] qualitative content analysis gave an opportunity to report data quantitatively as well. Due to data material, additional data exploration and analysis was made to explore results for patterns and co-occurrences. In addition to the individual unit of coding or category, the focus was on finding out how the categories were related. In linking the data and looking for connections or co-occurrences, frequency information was important and integrated into analysis. Comparing coding frequencies between the groups of sources was possible, at least to some extent. The three strategies used for presenting results in quantitative style were providing absolute frequencies, doing descriptive group comparisons, and using interferential statistics.

***The main categories and subcategories***—The main categories were named according to research questions and interview questions as (a) *the content of PE homework*, (b) *family and PE homework*, and (c) *doing PE homework*. Family included participants such as parents, mother, father, siblings (sister, brother), and even pets. The main categories included subcategories presented in Table 1. For example, *Family and PE homework* connection (b) could occur through family discussions, active participation, or some other reaction from family members. Subcategories were *told at home*, *with whom done with, where performed*, and *parent’s reaction*. The subcategory “*told at home*” applies when PE homework has been talked about with parents.

The parents’ interviews supplemented the students’ interviews, and the main category of parents’ interviews was predetermined as *family and PE homework.* The parent interviews were divided into five subcategories for the coding frame: *family members’ sporty background* (B), *parents’ attitudes towards school PE* (A), *PE homework as a familiar issue at home* (F), *parental support* (S), and *ideas or hopes for PE homework* (H) (Table 2). For example, *Family members’ sporty background* indicates parents’ sport activities and gave an expression of family’s attitude towards sport. The PE experiences and attitudes were placed in their own subcategory. In the category *concerning content of the PE homework or future ideas,* parents could mention several things, and all of them were marked separately, overlapping was intentional, and repletion was noted.

## 3. Results

### 3.1. Doing PE Homework

According to student interviews, talking about PE homework to family members was common; 28 students of the 38 expressed discussing PE homework at home (Table 1). The students usually told their parents about PE homework, and by far, parents held positive and accepting attitudes towards PE homework. However, PE homework assignments seemed to be a new and unfamiliar thing to parents. As an unknown phenomenon, PE homework could have caused resistance from parents, but these PE homework assignments seemed to slip into everyday lives effortlessly.


*“Yes, I have told at home that we have this kind of PE homework, and my parents were surprised and wondered what they are, because they’ve never had anything like that”.*

*(Minni 9th grade)*



*“I talk usually about PE homework at home, but our folks don’t participate”.*

*(Hertta 9th grade)*


According to parent interviews, their children had told them about PE homework (Table 2). One of the parents could not recall having been informed about the PE homework. However, he seemed to know about the homework because he had some critical views about burdening physically active girls with these PE homework assignments. Two of the parents had found out about the PE homework in some other way—for example, by seeing their children doing PE homework at home.


*“Yes, they have talked about those PE homework assignments, but there has not been any fuss about them”.*

*(Sari, parent)*



*“That is exactly that positive attitude towards physical activity at school, that PE homework, when it is voluntary, there is nothing negative in that”.*

*(Reija, parent)*


Even if the parents had not participated in doing PE homework, all of them had positive attitudes towards PE homework, as answers included 33 positive statements about PE homework. Two of the parents had critical points as well, the neutral attitude towards PE homework came up four times in total. One parent said that his children are more than physically active enough already, he criticized PE homework because of the extra burden it causes.


*“When I first heard about this PE homework, I thought what a great idea. Think about it, if it (PE homework) becomes a habit that would be amazing”.*

*(Reija, parent)*



*“For example, if you have a less active child, so that (PE homework) might motivate them being more physically active. Obviously physical activity has its’ benefits. But then the disadvantage, for example, my older daughters spend already so much time in their hobbies that then there should be enough time to do the homework and prepare for the exams and all…”*

*(Ari, parent)*



*“You should be able to just enjoy physical activity. When you are too goal-oriented it might get depressing”.*

*(Veera, parent)*


Parents realized the benefits of PE homework especially for less active adolescents. However, they also thought it was important to keep PE homework assignments fun and voluntary, not too goal-oriented, otherwise continuity would be compromised. As parents seem to appreciate the homework’s voluntariness, according to the self-determination theory, autonomy is a focal factor for motivation to participate in physical activities [30].

### 3.2. With Whom PE Homework Was Done with

A large number of the interviewees said that they usually did the PE homework assignments alone (18/38). In Finland, homework assignments in other subjects are so common and designed exactly to the right level that it is expected that adolescents can do them by themselves. Obviously, parents are expected to help when needed. However, 13 of the students expressed doing PE homework with the parents and 11 with their friends. Seven of the students did PE homework with siblings, four of them with a pet, and one student with sport teammate. 


*“My sister does them with me sometimes, and father joins if it’s jogging or something like that…or a dog”.*

*(Tanja 9th grade)*



*“These older ones take care of their homework independently anyway”.*

*(Ari, parent)*



*“Yes, I have talked about them at home, they have been surprised because they have never had anything like PE homework, but it’s fun to have to do some active tasks at home as well, it adds own physical activity…and with mom we have had some walks and so on, folks at home has to participate as well”.*

*(Miina, 9th grade)*


All interviewed parents currently had some physically active hobby, and their children also had organized physical activities. Two of the parents expressed supporting PE homework by participating in it, with six mentions in total. Altogether, finding time to do assignments together with a family member or a friend might be a challenge, it might just be easier to do the assignment alone. Assignments such as jogging or stretching seemed to be exercises which parents were eager to join. Those are common exercises that do not need any special skills or arrangements. Teaching the squatting technique was popular as well, this assignment included the situation where students had an opportunity to teach a sport skill to their parents.


*“*
*One assignment I remember especially. It was that doing squats up hill. It became a challenge, which one is going to be first at the top. It was funny for me as well, that is why it stick into my mind”. *

*(Reija, parent)*



*“Some kind of stretching tasks we have done together…I’m not sure if it was PE homework assignment or did the girls just want to activate their parents for some other reason”.*

*(Niina, parent)*


Parents appreciated PE homework assignments, where performing it with a family member was required. Three statements suggested that in the future, PE homework could consist of assignments that require doing them together. One parent expressed that on weekends, they do physical activities as a family; he mentioned trekking and swimming as their common activities.


*“It would be nice to participate with whole family, that way the assignment would have kind of double compulsory aspect, the assignment and involving parents”.*

*(Veera, parent)*



*“And when we go swimming, children are eager to go with a family. Otherwise, it is difficult to persuade them to go sporting with a family nowadays. They all have their own interests”.*

*(Ari, parent)*


Involving families and including the home environment in physical active assignments seem to be a desirable method, which might lead to adopting lifelong habits of physical activity. However, adolescents might no longer want to be physically active with their parents, and their interests might differ from their parents’ interests, as one father described in this study. As shown in this study as well, adolescents start to prefer self-initiated activities and leisure activities with their peers. In PE homework assignments, this was taken into account by giving students freedom to choose their company (parent, sibling, friend, dog) in doing required tasks.

### 3.3. Where PE Homework Assignments Were Performed

According to students, the two most popular places to perform PE homework were nearby neighborhoods (24) and home (23). Four of the students mentioned doing PE homework by incorporating it into organized sport activities. The alternative “somewhere else” had five mentions; these places were at school, commuting, or in sport facilities. Parents mentioned nature sports, but household chores were popular as well. The low-cost options, easy to achieve places, and familiar environments for physical activity were important for adolescents, self-initiated activities especially took place in a neighborhood setting or at home.


*“I usually do them at home, sometimes with my friends in our neighborhood”.*

*(Sofia 7th grade)*



*“Muscle training assignments I have done at home, and jogging stuff outside close to our home”.*

*(Elina 8th grade)*


In parents’ answers, nearby environment as a place where PE homework could be performed was mentioned often (nine statements). These students live in suburbs or in rural areas, they have a great opportunity for nature sport nearby. There are easily accessible sports fields, playgrounds, paths, and tracks as well. In Finland it is safe for children and adolescents to play, move, and commute by themselves [52,53].


*“And then something done in outdoor sport facilities, such as street workout equipment or track ‘n field exercises. Something that does not require special equipment”. *

*(Reija, parent)*



*“Activities like jogging or trekking in the forest, anything like that”.*

*(Ari, parent)*



*“There is a lot you can do at home as well”.*

*(Niina, parent)*


### 3.4. The Facilitators and the Barriers

For the students, the most frequently mentioned facilitator was fun assignments (12/38). Almost as popular was the fact that doing PE homework is recognized by the teacher somehow (11/38), for example, taken into account in evaluation. Valued aspects of PE homework were motivations for adolescents in PA (8/38), including the level of assignments being suitable and feeling good about doing the assignments (5/38). Three students mentioned the benefit that PE homework assignments could be done without any equipment, and three students felt they were motivated by accelerometer measurements performed for the research. As fun assignments, students mentioned brushing teeth, balancing on one leg, taking someone jogging with you, teaching the squatting technique to someone, using your “wrong” hand for tasks during the week, and fun and relaxing holiday homework assignments. Along with having fun as a main physical activity facilitator, perceptions of competence, acknowledgement of others, and motivation are proposed as well. For example, one student commented: *“I liked it, because I was good at it”.* The feedback point of view came from three students: two of them appreciated the plus marking in school intranet pages, and one mentioned the effect in evaluation and PE number. Students also mentioned that PE homework assignments do not take lot of time; most students (27/38) thought that PE homework should be voluntary, six of them expressed that they should be mandatory, and five were balanced with both options.


*“And PE homework is much more fun than homework from other subjects”.*

*(Alisa 7th grade)*



*“If doing PE homework could have an effect on PE number, and it is that feeling of accomplishment after doing them as well”.*

*(Natalia 9th grade)*


The students’ reasons for not doing the PE homework included forgetting it (10/38), being sick or convalescent (9/38), being too busy (5/38), not being motivated (5/38), or having too many organized sport activities already (4/38). Three mentioned the pressure in doing homework assignments as a barrier, as well as two having sore muscles. Not being a physically active person or not having the opportunity to do PE homework were mentioned by one student as well.


*“… and it’s good that they are voluntary, because if you are too busy or you have sore muscles or something then you don’t want start doing them”.*

*(Emily, 9th grade)*



*“There at school it must be challenging, because there is that whole group of students and they are comparing with each other”.*

*(Veera, parent)*


The parents thought that being active together at school might be challenging because of the possibly of being seen with your imperfections and comparing yourself to the others. Feeling uncomfortable in front of others or feeling incompetent regarding sports skills might create a barrier to physical activity for adolescent girls.

The most common statements from parents identifying good criteria for PE homework noted the versatile assignments. Parents valued the fact that no equipment or money was needed to perform PE homework and that the assignments were possible to perform in nearby environments. Mentions about easy enough assignments that anyone can manage came up frequently as well. The effect on evaluation and PE numbers and the assignments that enhance students’ fitness had several mentions; the voluntary nature of the assignments came up several times. Timeframe, household chores, utilization of technology, and humorous assignments were mentioned as well.


*“It is about thirty minutes that they can or bother to use (to PE homework), and if that is done like three times per week, it would be useful already”.*

*(Ari, parent)*


## 4. Discussion

In this study, we combined the views of students and parents, and provided a broad picture of the PE homework assignments that were given at school but completed at home. Both students and parents had positive attitudes towards PE homework assignments. Parents’ positive attitudes and support meant that they were interested in assignments, they acknowledged their children’s efforts in doing the assignments, and even arranged time to participate in doing them together. Even if parental support did not involve participating, parental cognitive engagement as supplementary to homework has been found as a positive type of parental involvement [54,55,56,57,58]. Higher levels of perceived support from parents and peers have been linked to adolescents’ enjoyment related to physical activity and beliefs regarding their motor skills, along with lower levels of perceived barriers in physical activity [20]. The method of parental support in homework depends on the achievement goals parents espouse for their children as well as the child’s own goal orientation [54,57,58]. Autonomy support has been found to be the most beneficial type of parental involvement in homework, whereas interference and parental control were found to be less adaptive [54]. Girls do not need parents to check their grades constantly and place pressure on girls who are already more likely to suffer from depression and stress in adolescence than boys [59]. Parents also wished that PE homework assignments would continue, and they had some specific visions regarding what PE homework assignments could look like in the future. Students’ positive attitudes were evident in talking about PE homework at home, highlighting the meaning of them, and actually participating in voluntary or extra PE homework assignments. Students also found more facilitators than barriers concerning the active PE homework. The feedback point of view came from both students and parents as they mentioned the effect in evaluation and PE number. At first, the feedback and grading might be good motivation for physical activity, but in the long term it is not a durable basis for lifelong habits [2,3,4,9]. However, acknowledgement and reinforcement by significant adults, may support students’ involvement in physical activities [59].

“Fun assignments” were the most popular facilitator for students in this research. Similar to that, having fun or experiencing enjoyment from physical activities were consistently mentioned as the facilitators or as a primary variable in participation of adolescent PA in several studies [4,5,9,20]. Students’ interviews revealed that in addition to fun assignments, they appreciated that their effort was acknowledged by influential adults, they found the motivation for PA, and the assignments were adequately challenging. This aligns with Martins et al.’s [5] findings, which suggested that key physical activity facilitators were motivation, perceptions of competence and fun, and the acknowledgement of significant others. The girls also appreciated the exercises that provided them with an outlet from daily routines, which was confirmed by parent interviews. Homework from other subjects might be taken more seriously, PE homework assignments were required to be designed to be inherently fun and not very goal oriented.

PE homework assignments involving family or parents’ participation were mentioned as a popular activity in both students’ and parents’ responses. However, doing PE homework alone was often preferred. That might reflect the typical habits of adolescents completing homework by themselves, as several of the interviewed parents mentioned in the study. Parents are involved in homework insofar as they believe their children and teachers want them to be, they think their involvement makes a difference, and they assume they have a role to play [60]. Teachers might help this involvement by communicating with parents about the importance of parental support, the role of participation, and what purpose homework is being used for, thus encouraging parents’ participation even with lesser skills [60,61,62]. Concerning homework, at higher student grade levels parents provide more homework autonomy and less attempts at help, parents might feel their skills are not sufficient and providing help interferes with their child’s studies [55].

Similar to prior research [4,5], the present study found that parents hoped PE homework assignments were diverse, able to be performed without equipment at home or in a nearby environment, and comparable with their child’s fitness level. In addition, physical activity in the school context should be aligned with the girls’ own preferences and promote their autonomy [5]. Following the national PE curriculum enables a low-cost way to provide versatile assignments as well as fostering perceptions of ownership [31,32,41,46,58]. In addition, PE homework assignments give teachers an opportunity to provide parents with a glance at the current contents and tasks of PE [47]. The assignments performed without any equipment at home or nearby the home enable being physically active regardless of students’ socio-economic backgrounds. Gill et al.’s [7] results indicated that students who have access to physical activity facilities or sport equipment are more likely physically active. Martins et al. [5] stated that adequate conditions are favorable to adolescents’ physical activity—for example, the existence of facilities and equipment and perceived neighborhood safety [7]. According to Kirby et al. [1], girls allowed to play outside without parents’ supervision were clearly more likely to be active. However, neighborhood safety did not come up from either parents’ or students’ answers, which was expected as participants were living in a safe, rural area of the small town in Finland [52,53].

The school day and especially PE lessons might be full of interaction and social collaboration; it is possible that some of the students might just appreciate alone time while doing PE homework. In addition, during adolescence changes in health behaviors, such as PA, become increasingly self-initiated decisions [4]. Homework assignments are meant to help students to develop their own skills and time-management and eventually become autonomous lifelong learners outside of formal educational settings [55,62,63].

Last, students in this study mentioned several barriers of PE homework, mainly the fact they forgot to do the assignments, or they were not well enough to perform it; being busy or no motivation were mentioned frequently as well, which aligns with prior research identifying the lack of time as the most consistent barrier to adolescents’ accumulation of sufficient physical activity [4,5]. In a study by Dias et al. [6], two obstacles most strongly associated with perceived barriers concerning leisure time physical activity: preferring to do other things and “feeling lazy”. “Lack of time” was reported as a barrier by almost half of the adolescent participants (14–18 years) as well. They suggest that helping to overcome the barriers could be a strategy for confronting adolescents’ physical inactivity and for the larger public health domain as well [6]. For example, there are useful technological solutions or help from social media to assist with remembering assignments and possibly obtaining immediate feedback on completion of the task. These technological solutions would likely facilitate this barrier for those adolescents who are having difficulties in remembering to complete active PE homework. Contrary to prior physical activity studies, in this study, however, time was not as significant of an issue for most of the participants as most mentioned that the PE homework did not take a lot of time.

### Limitations

Using cross-sectional data in one geographic location limits the generalizability of the results and prevents making causal claims. However, the Finnish national curriculum guides all education in Finland. Finnish PE is quite similar across the country, and results from one school might be adopted into other schools as well. Highly educated Finnish PE teachers can consider whether the families have the needed skills, time, and economic resources to participate in PE homework assignments [55].

Organizing the student interviews during their PE lessons has both benefits and drawbacks. The topic fits well in PE lessons, and it was feasible for both the interviewer and participants to organize interviews in the dressing rooms while the others were practicing. However, all the students who participated in sports did not want to have breaks during their PE practices for the interview session. Organizing student interviews during PE lessons might have had an effect on the study population.

All the interviewed parents described their sports backgrounds, and all interviewees’ children had some organized sport activities as well, causing study population bias of parents as well. Even if parents’ interviews revealed that physically active parents have physically active children [20], no generalizations can be drawn from this result [19,20]. The study group of parents was limited, the request via parent intranet resulted in only five voluntary parents. Because of the small sample size and selected sample containing only five voluntary parents and girls from one lower secondary school in Finland, the data cannot be considered representative, which could be a limitation and might preclude the detection of significance. However, these interviews provided a source of information that cannot be assessed by objective measurements. The interviews provided a rich context for understanding adolescents’ perceptions of PE homework, and parents’ perceptions supplemented students’ perceptions and made their opinion heard. This part of the PE Homework Study project focused on perceptions of parents and students, which supplements earlier parts of the study and the research body.

## 5. Conclusions

The idea and need for this study arose from everyday school life, and the experiment was carried out with normal students in an average school. The intention of this research was to provide evidence for much-needed improvements in the physical activity of adolescent girls. This research introduced the use of physically active physical education homework as an easily approachable tool to increase physical activity of adolescent girls after school hours and replace some sedentary activities with more active tasks. PE homework assignments should be fun, versatile, involve families, and have an impact on PE assessment. Both parents and female students appreciated family support in PE homework assignments. Due to that, diverse, interesting, and challenging physical activity opportunities should be provided with flexible schedules outside of school hours with family support in PE homework assignments.

The fact that adolescents’ physical activity is influenced by many factors indicates that to achieve any behavioral change, interventions must target several levels across the socio-ecological model. Since regular physical activity has positive impacts on health, and it improves the quality of life and overall well-being, identifying the factors that affect adolescent physical activity has public health significance as well. Schools have a unique position in our society, the vast majority of adolescents participate in school and physical education. This provides a great opportunity for low-cost school-based efforts and curriculum-based approaches for physical activity promotion in terms of public health and long-term health benefits, regardless of the participants’ social differences. PE homework assignments provide a tool for students to feel empowered and motivated to be physically active by offering physical activity opportunities on their own terms and choices, promoting feelings of autonomy and perceived motor competence with encouragement of the teacher and family. When the teacher’s promotion, encouragement, and interactions reach students in their daily life, in the form, for example, of physical education homework assignments, it might turn sedentary behavior into more active behavior and increase the physical activity of students. Active physical education homework assignment experiments in digital form might help students, and this kind of platform might be helpful in cooperation with teachers and schools as well. It might be time to expand the notion of physical activity and broaden the typical types of interventions that are provided to enhance activity in female adolescents. This study appears to hold promise as one such intervention.

## Figures and Tables

**Figure 1 ijerph-19-11924-f001:**
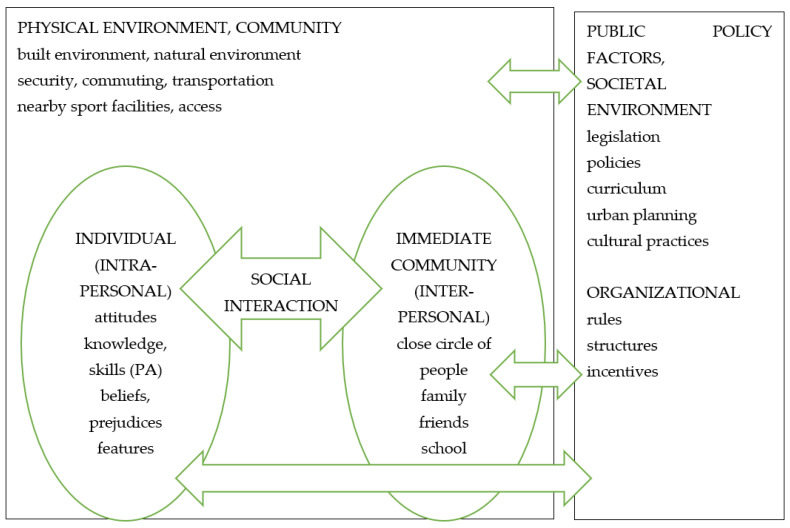
Interaction between the different levels of the socio-ecological model, framework modified according to Bronfenbrenner [48,49] and Elder et al. [12].

**Table 1 ijerph-19-11924-t001:** The analysis summary of students’ interviews (N = 38).

Subcategory	Code	Indicators	Statements
Told at home (G)	Ga	usually	26
	Gb	sometimes	2
	Gc	never	6
		Total	34
With whom done with (H)	Ha	parents	13
	Hb	friends	11
	Hc	siblings	7
	Hd	pets	4
	He	teammates	1
	Hf	alone	18
		Total	54
Parent’s reaction (N)	Na	positive	6
	Nb	negative	2
		Total	8
Where performed (F)	Fa	at home	23
	Fb	nearby environment	24
	Fc	along sport activities	4
	Fd	somewhere else	5
		Total	56
Barriers (M)	Ma	forgot	10
	Mb	sick/not healthy enough	9
	Mc	too busy	5
	Md	other hobbies	4
	Me	sore muscles	2
	Mf	no motivation	5
	Mg	pressure	3
	Mh	not sportive	1
	Mi	not possible	1
	Mj	lazy	1
		Total	41
Facilitators (L)	La	plus, marking	2
	Lb	effects PE number	1
	Lc	considered	11
	Ld	fun	12
	Le	good feeling	5
	Lf	measuring	3
	Lg	no equipment needed	3
	Lh	adequate level	7
	Li	timeframe	1
	Lj	others as well	2
	Lk	PA motivation	8
		Total	55
Participating (D)	Da	voluntary	27
	Db	mandatory	6
	Dab	both or not either one	5
		Total	38

**Table 2 ijerph-19-11924-t002:** The analysis summary of parents’ interviews (N = 5).

Subcategory	Indicators	Parents to Mention Statement/ All (Total Number of Mentions)
Family’s sportive background B	(a) parents have had sportive activities	Ba 4/5 (5)
	(b) parents have not had sportive activities	Bb 2/5 (2)
	(c) parents have sportive activities nowadays	Bc 5/5 (12)
	(d) parents do not have sportive activities	Bd 0/5 (0)
	(e) family participate on sportive activities together	Be 1/5 (1)
	(f) children have sportive activities	Bf 5/5 (5)
	Total	25
Attitude towards PE (school) A	(a) attitude towards PE positive	Aa 3/5 (4)
	(b) attitude towards PE negative	Ab 1/5 (4)
	(c) attitude towards PE neutral, not positive nor negative	
	Total	8
Do parents know about PE homework? F	(a) PE homework has been told or discussed about with parent/parents	Fa 3/5 (8)
	(b) parent/parents do not have the knowledge about the PE homework assignments	Fb 1/5 (1)
	(c) PE homework assignments have been revealed in some other way (seen children doing them, some other conversation revealed it…)	Fc 2/5 (5)
	Total	14
Family support to PE homework S	(a) PE homework has been done together at some point	Sa 2/5 (6)
	(b) positive attitude towards PE homework	Sb 5/5 (33)
	(c) negative attitude towards PE homework	Sc 2/5 (2)
	(d) supported some other way (transport, equipment, fees)	Sd 3/5 (4)
	(e) no support, not against it, neutral attitude towards PE homework	
	Total	45
What PE homework should be like? H	(a) should have an effect to PE number	Ha 4/5 (5)
	(b) easy enough, anyone can manage, not too difficult or hard	Hb 4/5 (8)
	(c) no equipment or payment needed	Hc 3/5 (9)
	(d) nearby environment used	Hd 3/5 (9)
	(e) timeframe mention	He 1/5 (2)
	(f) versatility	Hf 4/5 (10)
	(g) daily chores	Hg 1/5 (1)
	(h) utilization of technology	Hh 1/5 (1)
	(i) humoristic, funny, not so serious	Hi 1/5 (2)
	(j) improves the physical condition	Hj 2/5 (5)
	(k) done together with someone	Hk 2/5 (3)
	(l) voluntary	Hl 3/5 (4)
	Total	59

## Data Availability

All data are available from the corresponding author on reasonable request.

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
