# Peer review of "The Role of Parental Support and the Students’ Opinions in Active Finnish Physical Education Homework"

_ijerph, 2022, doi:10.3390/ijerph191911924_

Round 1
Reviewer 1 Report
This is well conceptualized and well written study. The only suggested revision include:
(1) Readers no doubt will be familiar with school-based physical education programs in their city, county, state, etc..Many if not most non-Finnish readers will not be familiar with school-based physical education programs in Finland. So to help better contextualize the study, it would be helpful to have more background on Physical Education in Finland. Is it required K-12? Who teaches physical education (degreed, licensed, etc.) teachers? Inform the readers of what physical education looks like in Finland
(2) Add additional narrative in the Conclusion section that expands on the implications of this study to public health. What might some of those health benefits be? Why should the public health profession embrace/support school-based physical education?
Author Response
Thank you for your excellent remarks, here are the revisions that I made:
1) A short information about physical education in Finland was added.
"
The Finnish National Curriculum, which is followed throughout the country, defines the boundary conditions and learning objectives for physical education [23]. All the teachers have university master’s level education, classroom teachers operate in grades from 1 to 6, and subject teachers at the lower secondary school (grades 7–9). In Finland, the distribution of physical education lesson hours during lower secondary school is seven hours weekly per year for three school years, meaning two or three hours per week during grades 7,8 and 9. The most common way is to organize 90 minutes (2x45 minutes) of physical education weekly. In addition, students have the opportunity to choose optional physical education which adds weekly amount of physical education. The school day often consists of 45-minute lessons and 15-minute breaks, which provides opportunities to be active during recess as well.
School sports contribute only marginally to adolescent’s physical activity, even well-organized physical education lessons do not exert a sufficient influence over physical activity and the health of adolescents [24–27]. To meet adequate levels of total physical activity, adolescent need to participate in physical activity outside of the school [28], the Finnish National Core Curriculum allows and encourages the use of leisure time to practice skills learned at school [23]. One strength of the Finnish educational system is that teachers have great deal of autonomy in deciding the way they teach [29]. Teacher can create opportunities for decision-making and regulate the amount of student involvement in it. Teacher can improve students’ autonomy and self-determination by involving them in planning and decision-making, choosing scale and complexity, or amount of physical active assignments [30]. By fostering perceptions of ownership over participation also in physical education and physical activity, students are enabled to be active in a way they choose [30– 32]."
2) The conclusion section was revised and the implications of this study to public health was added.
"Since regular physical activity has positive impacts on health, and it improves the quality of life and overall well-being, identifying the factors that affect adolescent physical activity has public health significance as well. Schools have a unique position in our society, the vast majority of adolescent participate in school and physical education. This provides a great opportunity for low-cost school-based efforts and curriculum-based approaches for physical activity promotion in terms of public health and long-term health benefits, regardless of the participants’ social differences."
I hope these additions will improve the manuscript sufficiently.
Reviewer 2 Report
Initially, I suggest that authors introduce the research objective in the Abstract.
The content of the introduction, method and results are well described and discussed.
In conclusion, I suggest that the authors present the limitations of the study, as only 34 young people were surveyed, as well as make explicit suggestions for future work.
I suggest minor revision.
Author Response
Thank you for your excellent suggestions, here are the changes/additions I made:
1) The research objective is introduced more clearly in the Abstract.
"The goal was to develop one easily approachable way to prevent the decreasing physical activity of adolescent girls. This was done by increasing physical activity time of adolescent girls outside of the school by giving them active PE assignments. The aim was also to explore students’ and their parents’ perceptions of physically active physical education homework."
2) The limitation of small sample size was added.
"Because of the small sample size and selected sample containing only five voluntary parents and girls from one lower secondary school in Finland, the data cannot be considered representative, which could be limitation and might preclude the detection of significance. However, these interviews provided a source of information that cannot be assessed by objective measurements. The interviews provided a rich context for understanding adolescents’ perceptions of PE homework, parents’ perceptions supplemented students’ perceptions and made their opinion heard. This part of the PE Homework Study project focused on perceptions of parents and students, which supplements earlier parts of the study and the research body."
3) The suggestions for future work.
"Experimenting active physical education homework assignments in digital form might help students, and this kind of platform might be helpful in cooperation with teachers and schools as well. "
I hope you will find these revisions adequate and sufficient.
Reviewer 3 Report
Abstract:
- the references should be removed from the abstract
- the abstract should be shortened, the introduction is too long
- conclusions should be added
Keywords:
keywords should not include abbreviations, such as PE homework
The introduction is detailed and the purpose is clear
Materials and Methods:
Study group: 38 students vs 5 parents. Please explain your choices. I see a big difference in the number of interviews. Why did only 5 parents take part in the study? It seems that 5 parents are not a good representation, not enough for the analysis
Please complete the ethics committee approval number
Results:
Authors shall be obliged to present only their results without citing the other authors.
Authors should also present the analysis of the results. The purpose of this article is to verify the thesis formulated in the title.
Presenting statements, on the other hand, is unnecessary.
Conclusions
the conclusions should be linked to the results of their own results, without citing the other authors
Author Response
Thank you for your excellent remarks of my manuscript, here are the revisions I have made concerning your suggestions:
1) The references were removed from the abstract.
2) The abstract, especially introduction part of it, was shortened.
3) Conclusions were added to the abstract part.
4) Keywords: PE was changed to physical education.
5) The limitation of small sample size was explained in Limitations part.
6) As I explained to editor, according to the Lawyer of Ethical Board, this research was accepted without further procedures, due to that, no approval number was given.
7) The results part was reformed and citations of other authors were reduced to the minimum. However, some essential citations were left to help the reader to understand the big picture and the context. The presenting statements was avoided as well.
8) The analysis is presented comprehensively (2.2 Qualitative Content Analysis), along with tables there are examples of coding, forming categories and subcategories, which rely on interview questions made to serve research questions. Authentic quotations reveal analysis process as well.
9) The conclusion part was rewritten, citations were removed and the conclusions were highlighted by the results of this study.
All the altered parts are marked with yellow color to the manuscript. I hope you will find these revisions adequate and sufficient.
Round 2
Reviewer 3 Report
The authors have addressed my concerns. Thank you for including the requested changes in the manuscript.